# OPEN-ENDED CONTENT-STYLE RECOMBINATION VIA LEAKAGE FILTERING

## ABSTRACT

We consider visual domains in which a class label specifies the *content* of an image, and class-irrelevant properties that differentiate instances constitute the *style*. We present a domain-independent method that permits the *open-ended* recombination of style of one image with the content of another. Open ended simply means that the method generalizes to style and content not present in the training data. The method starts by constructing a content embedding using an existing deep metric-learning technique. This trained content encoder is incorporated into a variational autoencoder (VAE), paired with a to-be-trained style encoder. The VAE reconstruction loss alone is inadequate to ensure a decomposition of the latent representation into style and content. Our method thus includes an auxiliary loss, *leakage filtering*, which ensures that no style information remaining in the content representation is used for reconstruction and vice versa. We synthesize novel images by decoding the style representation obtained from one image with the content representation from another. Using this method for data-set augmentation, we obtain state-of-the-art performance on few-shot learning tasks.

In any domain involving classification, entities are distinguished not only by class label but also by attributes orthogonal to class label. For example, if faces are classified by identity, within-class variation is due to lighting, pose, expression, hairstyle; if masterworks of art are classified by the painter, within-class variation is due to choice of subject matter. Following tradition (Tenenbaum & Freeman, 2000), we refer to between- and within-class variation as *content* and *style*, respectively. What constitutes content is defined with respect to a task. For example, in a face-recognition task, identity is the content; in an emotion-recognition task, expression is the content. There has been a wealth of research focused on decomposing content and style, with the promise that decompositions might provide insight into a domain or improve classification performance. Decompositions also allow for the synthesis of novel entities by recombining the content of one entity with the style of another. Recombinations are interesting as a creative exercise (e.g., transforming the musical composition of one artist in the style of another) or for data set augmentation.

We propose an approach to content-style decomposition and recombination. We refer to the method as STOC, for Style Transfer onto Open-Ended Content. Our approach is differentiated from past work in the following ways. First, STOC can transfer to novel content. In contrast, most previous work assumes the content classes in testing are the same as those in training. Second, STOC is general purpose and can be applied to any domain. In contrast, previous work includes approaches that leverage specific domain knowledge (e.g., human body pose). Third, STOC has an explicit objective, *leakage filtering*, designed to isolate content and style. No such explicit objective is found in most previous work, and as a result, synthesized examples may fail to preserve content as style is varied and vice versa. Fourth, STOC requires a labeling of entities by content class, but explicit style labels are not required. In contrast, some previous work assumes supervised training of both style and content representations.

Figure 1 shows examples of content-style recombination using STOC on the VGG-Face (Parkhi et al., 2015) data set. In each column, the content of the image in the top row is combined with the style of the image on the bottom row to synthesize a novel image, shown in the middle row. The images in the top and bottom rows are of identities (content) held out from the training set. Style is well maintained, and content is fairly well transferred, at least to the degree that the faces in the middle row are more similar to top-row than bottom-row faces. The training faces are labeled by identity, but style is induced by the training procedure.

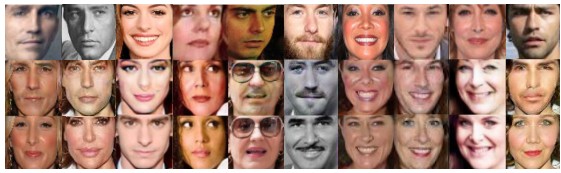

Figure 1: Examples of content-style recombination using STOC on the VGG-Face data set. The middle face in each column combines content of the top face with style of the bottom face.

# 1 PAST RESEARCH ON STYLE TRANSFER

A growing body of work has demonstrated impressive style transfer with models that can translate images from one specific domain (or content class) to another. Although some of these approaches require paired samples from both domains (Isola et al., 2017), several recent methods such as Cycle-GAN do not (Tzeng et al., 2017; Zhu et al., 2017; Choi et al., 2018). CycleGAN has been extended to exploit constraints in video (Bansal et al., 2018), yielding impressive sequences in which the mannerisms and facial movements of one individual are transferred to another. These methods are dependent on having *many* examples from pairs of content classes, and a model is custom trained for that pair. Therefore, the models do not attempt to learn an explicit representation of content or to decompose style and content.

Some domain-to-domain translation models do perform disentangling—of the information in an entity that is shared between domains and the information that is not shared (Gonzalez-Garcia et al., 2018; Huang et al., 2018). Huang et al. (2018) refer to this as content-style decomposition, but the range of content is quite restricted. For example, a model might be trained to transform cats into lions, but it cannot subsequently be used to transform cats into, say, panthers. An early proposal for style transfer (Kingma et al., 2014), based on variational autoencoders, can translate between more than two domains, but the model is unable to handle novel domains in the test set. Similarly, Structured GANs (Deng et al., 2017) can only be applied with a fixed set of classes. An open-ended method such as STOC can process novel content.

Many of the above techniques are described as unsupervised because no cross-domain correspondence between examples is required. However, from our perspective, the separation of examples by domain *is* a form of supervision, the same form we leverage in STOC.

Previous techniques that allow for open-ended content have typically required supervisory signals for both content and style. That is, labels must be provided for the content class of each training sample as well as for each of a specified set of style dimensions such as pose and lighting (Karaletsos et al., 2016; Kulkarni et al., 2015; Reed et al., 2014). Analogy constraints of the form $x_1 : x_2 :: y_1 : y_2$ have also been explored as a supervisory signal for style, specifying that two samples of one class $X$ have the same stylistic variation as two samples of another class, $Y$ (Reed et al., 2015).

Methods have been developed that can transfer style to novel content without requiring explicit style labels but instead rely on domain-specific knowledge. For example, Jetchev & Bergmann (2017) demonstrate the transfer of novel articles of clothing onto novel individuals, but their approach assumes that style transfer can be applied to only a masked region of the image. Other work has leveraged constraints inherent in a video sequence, either in a strong manner by extracting pose from the video (Brand & Hertzmann, 2000; Chan et al., 2018; Hsu et al., 2005), or in a weaker fashion by decomposing a video sequence into stationary (content) and nonstationary (style) components (Denton & Birodkar, 2017; Tulyakov et al., 2018). Neural Style Transfer and related methods (Gatys et al., 2016; Li et al., 2017; Wang et al., 2017) can do open-ended content-style recombination. However, it is limited in that it defines style as image *texture* (neural net features with a high degree of spatial correlation), and content as all other image features. While the method generates impressive results on texture transfer tasks such as translating a painting from one style to another, it is incapable of e.g., recombining faces with different pose, as shown in Figure 1.

# 2 OUR APPROACH

Our approach builds on a Variational Autoencoder (VAE) architecture (Kingma & Welling, 2013). We divide the latent code layer of the VAE into content and style components, as in the SSVAE (Kingma et al., 2014) and other recent work on probabilistic generative modeling (Siddharth et al.,

2017; Sohn et al., 2015). The content component is produced by a separately trained classifier, to be described shortly, which we will refer to as the *content encoder*. The style component uses the standard VAE encoding of posterior distributions over style vectors, with a prior determined by the variational loss. It is produced by a separate network called the *style encoder*. The content and (sampled) style serve as input to a *decoder* net, which synthesizes an image containing the two. The VAE reconstruction loss encourages the style vector to represent any additional input variability that cannot be attributed to class (content). Content-style recombination can be achieved in the obvious manner, by synthesizing an output that is based on content of one input and style of another.

We explore four variants of this model. The baseline model, which we refer to as CC for *content classifier*, uses a content encoder that is separately trained to be a one-hot classifier using a cross-entropy loss. This model cannot handle open-ended content because the training procedure requires data from all potential content classes. Nonetheless, it is useful as a reference point for comparison to other models. Our second variation uses a content encoder that produces an *embedding* rather than a one-hot encoding of class. The content encoder is trained with a deep metric learning objective, the histogram loss (Ustinova & Lempitsky, 2016), which has been shown to have state-of-the-art performance on few-shot learning (Scott et al., 2018). The embedding is $L_2$ normalized, in accordance with the fact that the histogram loss uses cosine distance. Because the content encoder produces a distributed representation of content, it can encode novel classes and is thus in principle adequate for handling open-ended content. We call this variation of the model CE for *content embedding*. Both CC and CE use the standard VAE loss, denoted $\mathcal{L}_{VAE}$. However, this loss does not explicitly disentangle content and style. Impurities—residual style information in the content representation and vice-versa—are problematic for content-style recombination. We thus propose two additional variations that add a *decomposition loss* aimed specifically at isolating content and style: *predictability minimization* (PM), which aims to orthogonalize representations, and *leakage filtering* (LF), which aims to filter out leaks and thereby obtain better style transfer.

## 2.1 PREDICTABILITY MINIMIZATION

Predictability minimization (Schmidhuber, 1992) encourages statistical independence between components of a representation via a loss that imposes a penalty if one component's activation can be predicted from the others. We apply this notion to style and content representations to minimize content predictability from style. (Because our content encoder is frozen when training the rest of the network, we do not implement the reverse constraint.) We build a content prediction net, or *CPN*, which attempts to predict, for training sample $\mathbf{x}$, the output of the content encoder, $z_{\mathbf{x}}^c$, from the output of the style encoder, $\{\boldsymbol{\mu}_{\mathbf{x}}^s, \boldsymbol{\sigma}_{\mathbf{x}}^s\}$. (The style encoder specifies the multivariate Gaussian style posterior obtained from the VAE.) Predictability minimization involves an adversarial loss:

$$\mathcal{L}_{PM} = \mathcal{L}_{VAE} + \lambda \min_{\theta_{CPN}} \max_{\theta_s} \mathbb{E}_{\mathbf{x} \sim X} ||z_{\mathbf{x}}^c - CPN(\boldsymbol{\mu}_{\mathbf{x}}^s, \boldsymbol{\sigma}_{\mathbf{x}}^s)||_2^2,$$

where $\theta_{CPN}$ and $\theta_s$ are parameters of the CPN and style encoder, respectively, and $\lambda$ is a scaling coefficient. Training proceeds much as in a generative adversarial network (Goodfellow et al., 2014).

## 2.2 LEAKAGE FILTERING

One way to ensure the success of style-content recombination is to remove all style information from $z_{\mathbf{x}}^c$ and to remove all content information from $z_{\mathbf{x}}^s \sim \mathcal{N}(\boldsymbol{\mu}_{\mathbf{x}}^s, \boldsymbol{\sigma}_{\mathbf{x}}^s)$. Another way is to simply ensure that the decoder filters out any leakage of content from $z_{\mathbf{x}}^s$ or leakage of style from $z_{\mathbf{x}}^c$ in forming the reconstruction. Leakage filtering (LF) achieves this alternative goal via constraints that guide the training of the decoder as well as the style encoder.

The constraints of leakage filtering are illustrated in Figure 2. In the left panel, we select a pair of samples of the same class, $\{\mathbf{x}, \mathbf{x}'\}$, from the complete set $P^+$, and use a decoder $D$ to recombine the style of $\mathbf{x}'$ with the content of $\mathbf{x}$ to synthesize an image $\mathbf{q}$. Because $\mathbf{x}$ and $\mathbf{x}'$ have the same content class, $\mathbf{q}$ should be identical to $\mathbf{x}'$. When they are not, style information may be leaking from $z_{\mathbf{x}}^c$. In the right panel, we select a pair of samples of different classes, $\{\mathbf{x}, \mathbf{y}\}$, from the complete set $P^-$, and transfer the style of $\mathbf{y}$ onto $\mathbf{x}$ to create a new image $\mathbf{r}$. Because $\mathbf{x}$ and $\mathbf{r}$ should share the same content, the content embeddings $z_{\mathbf{r}}^c$ and $z_{\mathbf{x}}^c$ should be similar; because $\mathbf{y}$ and $\mathbf{r}$ do not share the same content, $z_{\mathbf{r}}^c$ and $z_{\mathbf{y}}^c$ should be dissimilar. These constraints are violated when content information leaks from the style representation, $z_{\mathbf{y}}^s$. Just as the histogram loss was used to determine the content

embedding, we repurpose the loss to quantify the similarity/dissimilarity constraints in the content embedding. Here, however, the loss is used to adjust only parameters of decoder, $\theta_D$, and the style encoder, $\theta_s$.

The histogram loss is based on two sets of pairwise similarity scores, $S^+$ for pairs that should be similar and $S^-$ for pairs that should be dissimilar, as evaluated by a similarity function $s$; we use the cosine similarity. The histogram loss penalizes the overlap in the distributions of $S^+$ and $S^-$. We populate $S^+$ and $S^-$ with similarities of real-to-recombined samples as well as real-to-real, to ensure that the real-to-recombined similarities match the distributions of real-to-real:

$$S^+ = \left\{ s(z_{\mathbf{x}}^c, z_{\mathbf{x}'}^c), s(z_{\mathbf{x}}^c, z_{\mathbf{q}}^c) \mid \{\mathbf{x}, \mathbf{x}'\} \in P^+, \mathbf{q} = D(z_{\mathbf{x}}^c, z_{\mathbf{x}'}^s) \right\} \text{ and}$$

$$S^- = \left\{ s(z_{\mathbf{x}}^c, z_{\mathbf{y}}^c), s(z_{\mathbf{x}}^c, z_{\mathbf{r}}^c) \mid \{\mathbf{x}, \mathbf{y}\} \in P^-, \mathbf{r} = D(z_{\mathbf{x}}^c, z_{\mathbf{y}}^s) \right\}.$$

The histogram loss penalizes the overlap between $h^+(.)$ and $h^-(.)$, the empirical densities formed from the sets of similarity values in $S^+$ and $S^-$, respectively. The full LF loss is defined as:

$$\mathcal{L}_{LF} = \mathcal{L}_{VAE} + \lambda_1 \left( -\mathbb{E}_{\{\mathbf{x},\mathbf{x}'\} \in P^+, \mathbf{q}=D(z_{\mathbf{x}}^c, z_{\mathbf{x}'}^s)} \log \Pr[\mathbf{q} \mid \mathbf{x}'] \right) + \lambda_2 \left( \mathbb{E}_{s \sim h^-} \left[ \int_{-\infty}^s h^+(t)dt \right] \right),$$

where $\lambda_1$ and $\lambda_2$ are scaling coefficients. Because leakage filtering imposes a cost when the decoder fails to reconstruct an image, we have found the VAE reconstruction loss to be unnecessary. In the simulations we report, we replace $\mathcal{L}_{VAE}$ with $\mathcal{L}_{KL}$, the KL-divergence term of the VAE loss.

## 3 Experiments with Fixed Content

We begin with a data set having a fixed set of content classes, the MNIST handwritten digits (LeCun, 1998). Details of training, validation, and model architecture are presented in the Appendix. A qualitative comparison of content-style recombination of held-out test samples for CC, CE, PM, and LF variations is shown in Figure 3. In each case, loss weightings are hand tuned by visually inspecting recombinations from the validation set. In general, if too much weight is placed on reconstruction, the model will ignore content, and every row will look identical. If too much weight is placed on the decomposition loss or KL divergence, then there will be too much uniformity in a column, with little style transfer. In each grid of digits, the blue top row indicates the input digit (from the test set) used to specify content. The green leftmost column indicates the input digit used to specify style. Each gray digit is a sample from the network, with content specified by the corresponding blue digit, and style specified by the corresponding green digit. To the extent that content-style recombination is effective, all digits in a row should have the same style, all digits in a column should have the same class, and the two columns of each content should be identical despite variation in the blue digits. CC is superior to CE, but this result is unsurprising: representing content as a probability distribution over a fixed set of classes is a stronger constraint than a content embedding. Variant LF appears to be superior to either PM or CE, and surprisingly LF appears to be as good as, or better than, CC: the inductive bias of leakage filtering allows it to overcome the limitations of the weaker supervisory signal of the content embedding.

For a quantitative evaluation of the quality of synthetic digits, we investigate performance of a classifier trained from scratch on synthetic digits and tested on natural digits; we call this procedure *natural evaluation with synthetic training*, or *NEST*. If the synthetic digits do not look natural or have little stylistic variation, test performance is poor. To synthesize digits, we first select a *prototype*

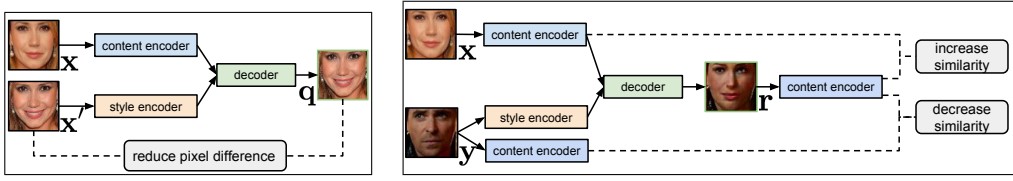

Figure 2: The logic of the leakage-filtering loss. Left panel: Leakage of style from the content embedding will cause $\mathbf{x}'$ and $\mathbf{q}$ to differ. Right panel: Leakage of content from the style embedding will cause $\mathbf{x}$ and $\mathbf{r}$ to have dissimilar content embeddings and $\mathbf{y}$ and $\mathbf{r}$ to have similar embeddings.

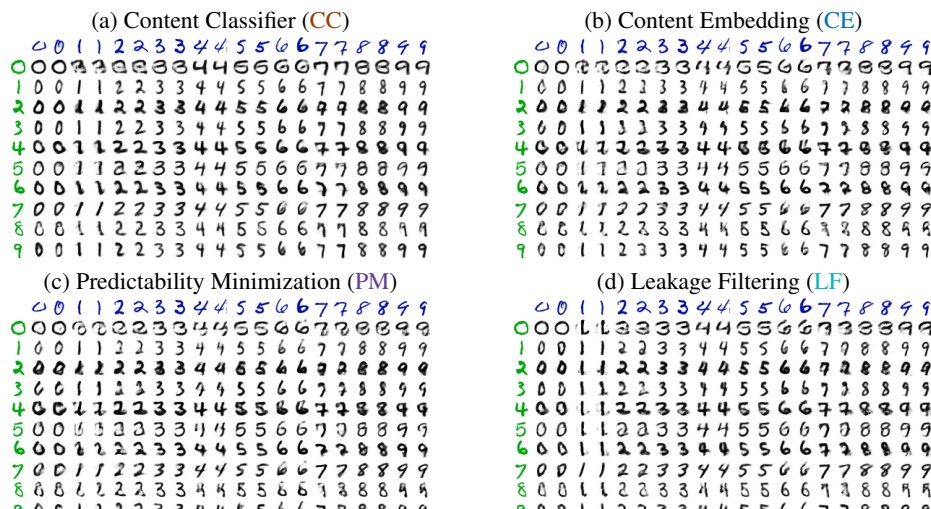

(a) Content Classifier (CC)

(b) Content Embedding (CE)

(c) Predictability Minimization (PM)

(d) Leakage Filtering (LF)

Figure 3: Content-style recombination on MNIST of alternative models. Black digits are synthesized from the style of green digit to the left and content of blue digit above.

content representation: The prototype content embedding for a digit class is the training instance that minimizes the sum squared Euclidean distance to all other instances of the same class. The prototype for CC is simply the one-hot vector for the given class. The classifier used for training has the same architecture as our content encoder, with 10 softmax outputs trained with a cross-entropy loss. Training is performed on minibatches of 40 samples with randomly-selected content and style provided by a random instance in (natural digit) training set, likely of a different class.

Figure 4 shows the mean probability of the correct class, a more sensitive metric than classification accuracy. Both PM and LF outperform the baseline CE, indicating that our losses to isolate content and style are doing the right thing. LF is clearly superior to PM, and in fact even beats CC, which is surprising because LF allows for open-ended content whereas CC does not. Because the VAE provides a prior over style, it is possible to simply sample style from the prior, rather than transferring it from another example. We repeated the NEST simulation using styles drawn from the prior and obtained similar results. Having shown the superiority of LF on a fixed set of classes, we next investigate performance of LF with open-ended content.

## 4    EXPERIMENTS WITH OPEN-ENDED CONTENT

We experiment with LF on two many-class data sets: Omniglot (Lake et al., 2015) and VGG Face (Parkhi et al., 2015). Details of data sets and split into training, validation, and test is in the Appendix. To improve the quality of our generated images in these more complex domains, we incorporate a WGAN-GP (Gulrajani et al., 2017) adversarial loss. This additional objective requires another scaling hyperparameter for the W-GAN loss, but training is otherwise identical to the MNIST procedure. We use a ResNet architecture for the content-style encoders, the decoder, and the critic

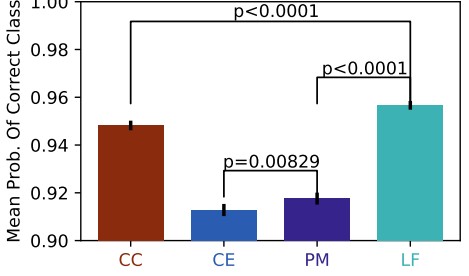

Figure 4: Naturally Evaluated, Synthetically Trained (NEST) results on MNIST. Mean probability of correct class is shown, with error bars indicating $\pm 1$ standard error of the mean. $p$ values are from two-tailed Bonferroni-corrected $t$-tests with 9999 degrees of freedom. All differences are highly reliable. CC = Content Classifier, CE = Content Embedding, PM = Predictability Minimization, LF = Leakage Filtering

Figure 5: Examples of recombined content and style using the Omniglot dataset of handwritten characters. The blue characters in the top row are test samples used to infer content, and the green characters are training samples used to infer style.

network of the WGAN-GP. For the VGG-Face data set, we include U-Net (Ronneberger et al., 2015) skip connections from both the style and content encoders to the decoder. Additional details can be found in the Appendix.

## 4.1 QUALITATIVE RESULTS

Figure 5 shows Omniglot characters with recombined content and style. Content is inferred from the blue character at the top of the column, a novel class from the test set. Style is inferred from the green character on the left, drawn from the training set. The content classes are repeated in order to determine how successful the model is at ignoring stylistic variation from the sample used to provide content. The three same-class digits in a given row are not always identical, but there is certainly more variation in a column (varying style) than there is in a row triplet (varying samples providing the content). All characters in a row appear to share stylistic features: e.g., they are very small, have wavy lines, are bold, or are boxy in shape.

Figure 6 shows examples of VGG Faces with recombined content (the face in the top row) and style (the face in the left column). Looking across a row, the model preserves many aspects of style, including pose, lighting conditions, and facial expression. In the last row, even glasses are considered a stylistic feature, surprising given the strong correlation of glasses presence across instance of an individual. Looking down a column, many identity-related features are preserved, including nose shape, eyebrow shape, and facial structures like strong cheekbones.

## 4.2 APPLICATION TO DATA AUGMENTATION

Next, we explore using STOC for data augmentation and evaluate on few-shot learning tasks. Data augmentation is the process of synthesizing variations of a training sample by transformations known to preserve some attribute of interest to a task (e.g., object class), in hopes that a predictive model will become invariant to the introduced variations. Domain-specific techniques are very common, especially in perceptual domains, e.g., image translation and flipping. Style transfer using STOC provides a domain-agnostic method.

Recently, other researchers have used machine learning to augment data. Several methods make use of generative adversarial nets to refine images produced by CAD programs (Shrivastava et al., 2017; Sixt et al., 2018), but these obviously rely on significant domain knowledge. DeVries & Taylor (2017) generate new samples of a class by interpolating the hidden representations of labeled samples of that same class. Zhu et al. (2018) generates augmented faces for emotion recognition. Emotion is defined as the content class and a CycleGAN-like architecture is used to translate from one emotional expression to another. This approach works only for a fixed set of known classes and therefore cannot be directly compared to STOC. Two papers (Rezende et al., 2016; Antoniou et al., 2017) introduce methods for generating new samples that share a class with a given input sample, and are shown to work with novel classes. Only Antoniou et al. (2017) demonstrates performance on a data-augmentation task, so we choose this paper as our primary point of comparison.

Because data augmentation should have the greatest effect in data-sparse domains, we evaluate STOC augmentation on few-shot learning, where the goal is to obtain accurate classification based on a small number of samples. Our evaluation procedure follows Scott et al. (2018). The data set is divided by content-class into source ($\mathcal{S}$) and target ($\mathcal{T}$) domains. $\mathcal{S}$ is split by class into a training

and validation set, used to train STOC. We use $\mathcal{T}$ for evaluation. Within $\mathcal{T}$, each class has $N$ samples, which are split into $k$ *support* samples (which together make up $\mathcal{T}_s$), and $(N - k)$ *query* samples ($\mathcal{T}_q$). Testing proceeds in *episodes*, where a subset of $n$ classes is drawn from $\mathcal{T}$ for testing. We then generate the augmented set ($\mathcal{T}_a$) using the content of $\mathcal{T}_s$, and style drawn from $\mathcal{S}$. A classifier is then trained using $\mathcal{T}_{sa} \equiv \{\mathcal{T}_s, \mathcal{T}_a\}$. Performance is reported on the classification accuracy of $\mathcal{T}_q$. We evaluate two different methodologies. First, we compare our method to other state-of-the-art one-shot learning methods on the Omniglot dataset. Second, we consider the case of training a new classifier from scratch on only $\mathcal{T}_{sa}$.

**One-Shot Learning with Omniglot**. We investigate the common one-shot Omniglot task, where the number of classes per episode ($n$) is 20, and the number of examples per class ($k$) is 1. To generate $\mathcal{T}_a$, we synthesize $m$ stylistic variations of each member of $\mathcal{T}_s$. We experiment with two settings, $m = 0$ (no augmentation) and $m = 40$. Also, we found that limiting the variability introduced by style transfer to be important, so instead of replacing the style of the samples of the support set with the style of a training example, we linearly interpolate between them.

Scott et al. (2018) demonstrated that the histogram-loss embedding achieves state-of-the-art performance on this task. We use the histogram embedding of the content-encoder network that is trained for STOC, ensuring that there is no performance difference between the content embedding used to train the style transfer model and the embedding used for few-shot learning. To evaluate an episode, we first embed the $\mathcal{T}_{sa}$ set using the content encoder. For each query sample, we compute its content embedding. We then compute the $L_2$ distances between the query embedding and each embedding in $\mathcal{T}_{sa}$. For each embedding in $\mathcal{T}_{sa}$, we assign a weight to determine the contribution strength of that sample to the overall decision. Each real support sample is assigned a weight $w_s$, and each of the $m$ augmented samples is assigned a weight $w_a = (1 - w_s)/m$. The probability distribution over classes is computed via a weighted softmax on the squared distance between the query sample and the samples in $\mathcal{T}_{sa}$.

For each episode, we record the average classification accuracy for all the query samples. We run 400 episodes, each with different random subsets of test classes, and report average accuracy across the replications. Table 1 shows the results for our model with and without data augmentation, along with reported results from the literature. For this task, we find that the baseline histogram performance is already very good. Although the improvement from data augmentation is small, it brings

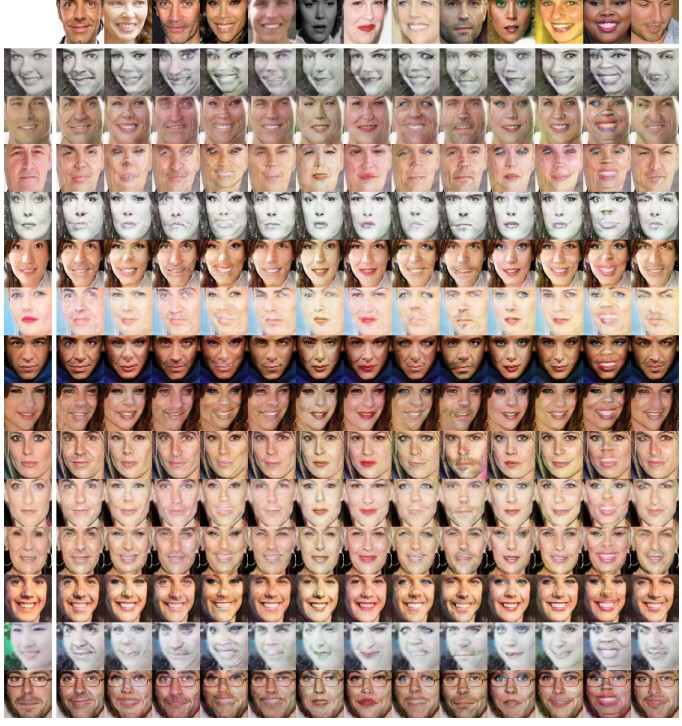

Figure 6: Examples of content-style recombination using the VGG-Face data set. The images in the matrix are formed by re-combining the content (identity) of the image in the top row with the style of the image in the left column. The top row contains samples from identities held out from training. The left column contains other samples from the data set.

| Model Name | Test Accuracy |
|---|---|
| Matching Nets (Vinyals et al., 2016) | 0.938 |
| Prototypical Networks (Snell et al., 2017) | 0.960 |
| Matching Nets (DAGAN replication) (Antoniou et al., 2017) | 0.969 |
| Matching Nets + DAGAN Augmentation (Antoniou et al., 2017) | 0.974 |
| Conv. ARC (Shyam et al., 2017) | **0.975** |
| Histogram Embedding (our implementation) (Ustinova & Lempitsky, 2016) | 0.974 |
| STOC (ours) | **0.975** |

Table 1: Average query accuracy for the one-shot learning task with the Omniglot data set, with $k = 1$ samples per class in the support set and $n = 20$ classes per episode.

the histogram embedding performance up to the level of Conv-ARC (Shyam et al., 2017), which is a complex, articulated, recurrent architecture with attention that performs explicit comparisons between samples. DAGAN (Antoniou et al., 2017) shows a bigger improvement, but it makes use of an auxiliary *sample-selection network*, the details of which are not explained.

**Standard Classifiers with Omniglot and VGG-Face**. We also trained standard classifiers from scratch on $\mathcal{T}_{sa}$. The classifiers are convolutional nets with 4 strided convolutional layers, followed by a ReLU activation, batch norm, and dropout with a rate of $0.5$. Each convolutional layer has a kernel size of 5 and 64 filters. To train the nets, we split $\mathcal{T}_s$ into training (75%) and validation (25%) sets, and use the validation set to determine the number of epochs to train for. Minibatches are composed of some mixture of real and augmented samples, and we used the validation set to determine the ratio. We generate new $\mathcal{T}_a$ augmentations for every minibatch. For omniglot, we also experiment with adding "standard" data augmentations (rotations, shifts, and dilations; see appendix for details). For VGG-Face, we do not add augmentations because the images have been already been carefully pre-processed to normalize face rotation, shift, and zoom. Table 2 shows the results on the test samples for both Omniglot and VGG-Face data sets. For Omniglot, we report results on the whole set of 1299 test classes, varying $k$, the number of samples per class in the support set. To compare our results with DAGAN (Antoniou et al., 2017), we select a random subset of the 212 Omniglot classes, which is the size of the DAGAN test set, and we use the same test set size as DAGAN for VGG Face. For omniglot, standard augmentation improves accuracy over baseline in every case, and the additional augmentation from STOC further improves performance. Likewise, for VGG-Face, STOC augmentations improve performance. We demonstrate that STOC performance on data augmentation is about on par with DAGAN, even though DAGAN was specifically designed for this task. STOC and DAGAN have different goals, and it is valuable to study both approaches for data augmentation. The fact that both models perform similarly might point to a limitation to the potential benefit of synthetic data for training.

| | | | STOC | | | Antoniou et al. (2017) | |
|---|---|---|---|---|---|---|---|
| | | | *Test Accuracy* | | | *Test Accuracy* | |
| *Data set* | *n* | *k* | *Baseline* | *Std. Aug.* | *Std. Aug. + STOC* | *Std. Aug.* | *Std. Aug. + DAGAN* |
| Omniglot | 1299 | 5 | 0.261 | 0.560 | 0.631 | | |
| | | 10 | 0.426 | 0.679 | 0.688 | | |
| | | 15 | 0.543 | 0.696 | 0.703 | | |
| Omniglot | 212 | 5 | 0.435 | 0.683 | 0.807 | 0.690 | 0.821 |
| | | 10 | 0.571 | 0.833 | 0.857 | 0.794 | 0.862 |
| | | 15 | 0.643 | 0.821 | 0.879 | 0.820 | 0.874 |
| VGG Face | 497 | 5 | 0.087 | | 0.272 | 0.045 | 0.126 |
| | | 15 | 0.263 | | 0.448 | 0.393 | 0.429 |
| | | 25 | 0.371 | | 0.504 | 0.580 | 0.585 |

Table 2: Results for training standard classifiers on un-augmented and on augmented data. The columns, from left to right: the tested data set, the number of classes in the training/test data sets ($n$), the number of samples per class in the support set ($k$), the baseline (un-augmented) test accuracy, the accuracy of a model with standard data augmentation, and the accuracy of a model with both standard data augmentation and STOC augmentation. We have also listed the test accuracy from Antoniou et al. (2017), where appropriate.

## 5 CONCLUSION

STOC is effective in transferring style onto open-ended content—content that is novel with respect to the training data. This is a challenging task: content class boundaries cannot be determined precisely in a setting where the number of potential classes is unbounded. As a result, it is easy for some style information to seep into a content representation. We introduced the leakage-filtering loss, a novel approach to isolating content and style. Traditionally, researchers have focused on disentangling style and content: inducing representations that separate style and content into different vector components. Given the difficulty of this challenge with only content labels and no explicit labels or domain knowledge pertaining to style, we instead focus on ensuring that the decoder, which combines style and content to reconstruct images, does not use any residual style information in the content representation or any residual content information in the style representation. Our results yield impressive visual quality and achieve significant boosts in performance when STOC is used for augmenting data sets to train a de novo classifier. We also explored data augmentation for few-shot learning and achieved performance that matches state of the art, a complex highly articulated and computation intensive model. We suspect that beating state-of-the-art on few-shot learning is becoming increasingly difficult, given that state-of-the-art is now bumping against the ceiling on performance in the paradigm that is typically used for evaluation.

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

# A APPENDIX

## A.1 DATA SETS

**MNIST**. The MNIST database is divided into a development set consisting of 60,000 images and a test set of 10,000. We further divide the development set into a training set of 48,000 and validation set of 12,000, which is used to determine model hyper-parameters such as the number of epochs to train, and relative weights on the various loss functions.

**Omniglot**. Omniglot is composed of 20 instances of 1,623 different classes of hand-written characters from 50 different alphabets. Following convention (Scott et al., 2018; Snell et al., 2017; Triantafillou et al., 2017), we augment the data set with all $90°$ rotations, resulting in 6,492 classes. The classes are split randomly into 4,154 training, 1,039 validation, and 1,299 test classes.

**VGG Faces**. We use the same subset of the data as (Antoniou et al., 2017), splitting the data into 1,750 training classes, 53 validation classes, and 497 test classes.

## A.2 NETWORK ARCHITECTURES AND PARAMETER SETTINGS

**Architecture for MNIST**. The content and style encoders have the same network architecture: two convolutional layers with $5 \times 5$ kernels and 64 filters, followed by a fully connected layer. Each convolutional layer includes batch normalization and leakly ReLU activation function. The three variations with content embeddings (CE, PM, LF) use the same content encoder network with 50 dimensions. Figure 7 shows a two-dimensional t-SNE visualization of the embedding of the content encoder, with four randomly sampled digits of each class from the test set. Some within-class variation is preserved, but the digit classes are still well separated. CC has a 10-dimensional (one hot) content representation. All four variations use 50 dimensions for the style representation. The content representation is $L_2$ normalized to have a unit length, required for the histogram loss. The generator network consists of a linear projection from the 100-dimensional combined content and style representation to a $6 \times 6 \times 32$ tensor. This tensor is then up-sampled twice using transposed, fractionally-strided convolutions, each with a kernel size of $5 \times 5$ and 64 filters. Finally, the output of the generator is normalized to lie within $[-1, 1]$ using a hyperbolic tangent activation. The inputs to the network are also normalized to lie in this range. All networks were trained with the Adam optimizer (Kingma & Ba, 2014) with a learning rate of $2 * 10^{-4}$. For predictability minimization, the CPN network consists of simple feed-forward net with one 100-dimensional hidden layer and a leaky ReLU activation.

**Architecture for omniglot**. For the content and style encoders, we start with a simple convolutional layer, followed by three ResNet blocks, with three convolutional layers each and a skip-connection at the end. We add a final convolutional layer to reduce the output channels to 3 (RGB). The output is normalized with a hyperbolic tangent function to lie within $[-1, 1]$. The content and style representations are both 100-dimensional. After each convolution, we use a leaky ReLU activation, and then batch re-normalization (Ioffe, 2017). Each convolutional layer has a $3 \times 3$ kernel and 48 filters. For the decoder and WGAN-GP critic, we use the an identical architecture, except that each

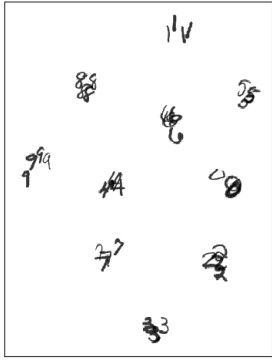

Figure 7: Example t-SNE visualization of the histogram embedding of MNIST digits.

convolutional layer has 64 filters. As noted in Gulrajani et al. (2017), batch normalization violates the assumptions of WGAN-GP, so our critic net uses layer normalization (Ba et al., 2016). The model was trained for 100 epochs, using the Adam optimizer and a learning rate of $10^{-4}$. We set the coefficient on the KL-divergence loss to 1, and the coefficients on $\lambda_1$ and $\lambda_2$ of $\mathcal{L}_{LF}$ to 20, and the WGAN-GP loss is multiplied by 0.5. We set the gradient penalty weight parameter of WGAN-GP to 10.

**Architecture for VGG-Faces**. We start with the same ResNet architecture that we used for omniglot, but use 64 filters in each convolutional layer of the style and content encoders. We also include U-Net skip connections from the output of each resnet block in both the content encoder and the style encoder to the corresponding ResNet block input in the decoder. The WGAP-GP critic's architecture is identical to that of the decoder, except it has no U-Net skip connections. The content representation has 200 dimensions, and the style representation has 600. The model was trained for 10 epochs, using the Adam optimizer and a learning rate of $10^{-4}$. We set the coefficient on the KL-divergence loss to 1, and both of the coefficients on $\lambda_1$ and $\lambda_2$ of $\mathcal{L}_{LF}$ to 5, and the WGAN-GP loss is multiplied by 0.5. For VGG-Faces, we find a higher value of the weight on the gradient penalty parameter (100) gives better training stability.

**Minibatch Composition**. For MNIST, minibatches were constructed of 4 samples of each of the 10 classes, with every possible positive- and negative comparison included in the $\mathcal{L}_{LF}$ objective. To train the content encoders for both omniglot and VGG-Faces, we construct minibatches by sampling 20 classes, with 10 samples each, which is close to the recommended batch size of 256 from Ustinova & Lempitsky (2016). For performance reasons, we sample only 10 classes with 3 samples each to train the style encoder, generator, and WGAN-GP critic. We sub-sample the between-class comparisons such that their count equals the count of within-class comparisons.

**Parameters for One-Shot Learning**. We find that setting the weight on the real support data $w_S$ to a value around 0.85 to give the best results. We also found that a relatively low temperature in the softmax fuction works better than a high temperature (we used $T = 0.05$).

**Standard Data Augmentation for Training Classifiers on Omniglot** We selected three different class-preserving standard augmentations for omniglot. We added random rotations between -20 and 20 degrees and random shifts between -3 and 3 pixels in both vertical and horizontal directions. We also added croppings of the $28 \times 28$ original images to a random size as small as $25 \times 25$, and rescaled the cropped image back to $28 \times 28$. The amount of standard augmentation was selected based on the validation set.

## A.3 EXPLORATION OF LOSS COEFFICIENTS ON MNIST

Figure 8 shows the effect of varying the weights on reconstruction and decomposition losses on performance on the Naturally Evaluated / Synthetically Trained task, using the validation set. In all cases, the KL-divergence loss $\mathcal{L}_{KL}$ was set to a constant (1). The effect of the weight on reconstruction loss is shown in Figure 8-(a). When the weight is too small, reconstructions become blurry and the generated digits are not useful samples for training. When the weight is too large, the network learns to ignore content, reconstructing only from style. In this case, the generated images look better, but not of the intended class. The blue line in Figure 8-(b) shows the effect of the weight on predictability minimization, with the reconstruction weight clamped to X. The green dotted line shows performance with no predictability minimization. When the weight is too small, we hypothesize that predictability minimization can still interfere with the network's ability to reconstruct, but without providing much benefit in terms of reducing representation redundancy. When the weight is too large, the network can no longer generate good images. Figure 8-(c) shows the effect of both coefficients of $\mathcal{L}_{LF}$ ($\lambda_1$ and $\lambda_2$) on NEST performance. Since there are two coefficients, we demonstrate the effect of one when the other is held constant at its best setting. The left-hand plot shows the effect of $\lambda_1$, which governs the component that filters leakage of style from $z^c$ when generating an image. When this parameter is set too low, reconstructions are blurry and not useful for training. We found that as long as this parameter is set to a high enough value, increasing it further does not appear to affect the quality of generated images. The righthand plot of Figure 8-(c) shows the effect of $\lambda_2$, which governs the component that filters leakage of content from $z^s$ when generating an image. Again, we find that as long as this parameter is not set too low, performance is relatively

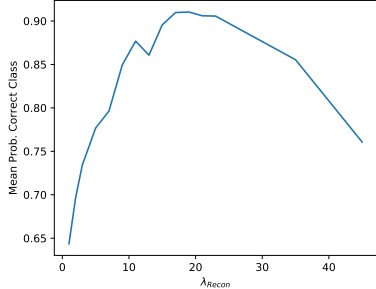

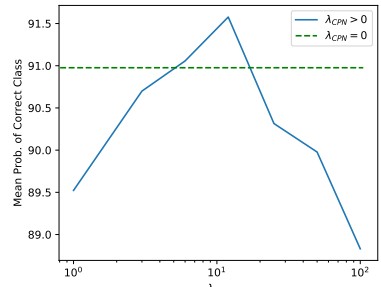

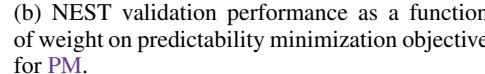

(a) NEST validation performance as a function of weight on reconstruction objective for CE.

(b) NEST validation performance as a function of weight on predictability minimization objective for PM.

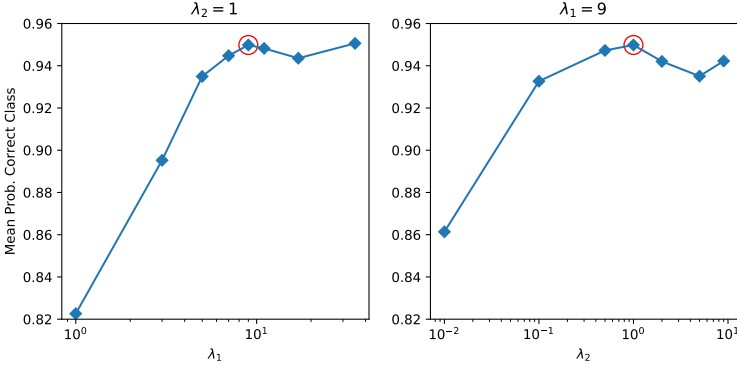

(c) NEST validation performance as a function of weights on LF. The best-performing model is highlighted in red.

Figure 8: Explorations of loss weights for CE, PM, and LF.

insensitive to its value. This behavior is advantageous for LF: its insensitivity to relative weighting means that not much hand-tuning is necessary, while CC and PM need a great deal of hand tuning.

## A.4    EFFECT OF PREDICTABILITY MINIMIZATION ON CONTENT INFORMATION IN STYLE

If predictability minimization is working properly, it should reduce the amount of content information recoverable from the style representation. To investigate the effect, we trained several STOC models on MNIST using predictability minimization, and varied the weight on the PM loss. After training, we train feedforward neural net with one hidden layer that predicts digit-class from the posterior mean and variance of the style representation. As content information is removed from style, the classifier should be increasingly inaccurate. The blue line in Figure 9 shows style-conditional content classification accuracy as a function of the weight on the predictability minimization decomposition loss. The dotted green line shows the accuracy when predictability minimization is disabled. As expected, the accuracy decreases with greater weight on PM.

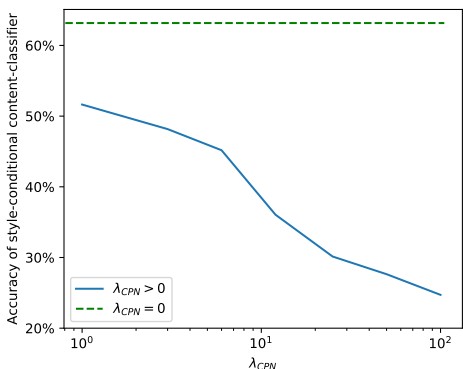

Figure 9: Accuracy of a post-hoc network trained to predict content from style, as a function of weight on Content Prediction Net (CPN) objective of PM.

