# OpenReview forum: "Open-Ended Content-Style Recombination Via Leakage Filtering"
_ICLR.cc/2019/Conference_

### Official Review · AnonReviewer1 · 2018-10-31

**Rating:** 5
**Confidence:** 4

**Review:**

UPDATE:
Thanks for your response. As you mentioned, methods like [1] and [2] do perform open-ended recombination. Note that these methods perform not only texture transfer but also color transfer, while the proposed method seems to perform mostly only color transfer. As shown in Figure 6, essentially what the method does is transfer the color of the style image to the content image, sometimes with a little tweak, making the image distorted. One could say that in terms of image style transfer, the proposed method actually underperforms [1] and [2].

Hence I agree with R2 that comparison is still necessary for the submission to be more convincing and complete.

------------------------------

This paper proposed to use a mechanism of leakage filtering to separate styles and content in the VAE encoding, and consequently enable open-ended content-style recombination. Essentially the model tries to maximize the similarity between images in S^+ and minimize the similarity between those in S^-.

I have several questions:

One concern that I have is the relationship/difference between this work and previous work on style transfer, especially universal/zero-shot style transfer as in [1,2]. In the introduction and related work sections, the authors argue that most previous work assumes that content classes in testing are the same as those in training, and that they are not general purpose. Note that various works on style transfer already address this issue, for example in [1, 2]. For those models, content is represented by high-level feature maps in neural networks, and style is represented by the Gram matrix of the feature maps. The trained model is actually universal (invariant to content and styles). Actually these methods use even less supervision than STOC since they do not require labels (e.g., digit labels in MNIST).

This brings me to my second concern on proper baselines. Given the fact that previous universal/zero-shot style transfer models focus on similar tasks, it seems necessary to compare STOC to them and see what the advantages of STOC is. Similar experiments can be conducted for the data augmentation tasks.

In Sec. 4, the authors mentioned that U-Net skip connection is used. Does it affect the effectiveness of the content/style separation, since the LF objective function is mostly based on the encoding z, which is supposed ‘skipped’ in STOC. Will this lead to additional information leakage?

It is not clear how the last term of L_{LF} is computed. Could you provide more details?

The organization and layout of figures could be improved. The title/number for the first section is missing.

Missing references:

[1] Universal style transfer via feature transforms, 2017
[2] ZM-Net: Real-time zero-shot image manipulation network, 2017
[3] Structured GAN, 2017

---

> ### Author Response · Authors · 2018-11-26
> **Response**
>
> We thank the reviewer for their careful evaluation and feedback. Regarding related work you cite, your references [1] and [2] are variations on the neural style transfer method of Gatys et al. [2016]. In this approach, content is defined as the image itself, and style is defined in terms of spatially invariant image statistics (the Gram matrix). This model is often described as a texture-transfer method, although the notion of texture can be quite abstract. Although this method does perform open-ended recombination (it can work for any new pair of images defining content and style), it is limited to transferring texture and not arbitrary style. For example, it could not re-render faces in different poses as in our Figure 1. It’s therefore highly unlikely to work well for tasks such as face or character style transfer. The structured GAN--your reference [3]--is shown to work only with a fixed set of classes. We appreciate that the method could be extended use an embedding representation to work with open-ended content, but this extension is beyond the scope of work for a comparison. We have incorporated a discussion of the related work [1,2,3] in in our manuscript.
>
> Regarding the U-Net skip connections and their effect on the leakage filtering objective: Leakage filtering places constraints on the recombined images, rather than on the latent representations. It is therefore compatible with architectures using skip connections. Predictability minimization is a regularizer on the latent representation, and would be incompatible with skip connections, but we do not explore that case due to the poor performance of predictability minimization.
>
> Regarding how the final term of the leakage filtering loss, $L_{LF}$ is computed: the histogram loss [Ustinova et al., 2016] is used for evaluating the content in a reconstruction, exactly as we used the histogram loss for determining the content embedding in the first place.  It is a simple and elegant approach.

---

### Official Review · AnonReviewer2 · 2018-11-03
**Well-written, but incomplete comparisons to prior work**

**Rating:** 5
**Confidence:** 3

**Review:**

SUMMARY
The paper considers several methods for building generative models that disentangle image content (category label) and style (within-category variation). Experiments on MNIST, Omniglot, and VGG-Faces demonstrate that the proposed methods can learn to generate images combining the style of one image and the content of another. The proposed method is also used as a form of learned data augmentation, where it improves one-shot and low-shot learning on Omniglot.

Pros:
- The paper is well-written and easy to follow
- The proposed methods CC, CE, PM, and LF are all simple and intuitive
- Improving low-shot learning via generative models is an interesting and important direction

Cons:
- No comparison to prior work on generation results
- Limited discussion comparing the proposed methods to other published alternatives
- No ablations on Omniglot or VGG-Faces generation
- Low-shot results are not very convincing

COMPARISON WITH PRIOR WORK
There have been many methods that propose various forms of conditional image generation in generative models, such as conditional VAEs in [Sohn et al, 2015]; there have also been previous methods such as [Siddharth et al, 2017] which disentangle style and content using the same sort of supervision as in this paper. Given the extensive prior work on generative models I was a bit surprised to see no comparisons of images generated with the proposed method against those generated by previously proposed methods. Without such comparisons it is difficult to judge the significance of the qualitative results in Figures 3, 5, and 6. In Figure 3 I also find it difficult to tell whether there are any substantial differences between the four proposed methods.

The proposed predictiability minimization is very related to some recent approaches for domain transfer such as [Tzeng et al, 2017]; I would have liked to see a more detailed discussion of how the proposed methods relate to others.

OMNIGLOT / VGG-FACES ABLATIONS
The final model includes several components - the KL divergence term from the VAE, two terms from LF, and a WGAN-GP adversarial loss. How much do each of these terms contribute the quality of the generated results?

LOW-SHOT RESULTS
I appreciate low-shot learning as a testbed for this sort of disentangled image generation, but unfortunately the experimental results are not very convincing. For one-shot performance on Omniglot, the baseline Histogram Embedding methods achieves 0.974 accuracy which improves to 0.975 using STOC. Is such a small improvement significant, or can it be explained due to variance in other factors (random initializations, hyperparameters, etc)?

For low-shot learning on Omniglot, the proposed method is outperformed by [Antoniou et al, 2017] at all values of k. More importantly, I’m concerned that the comparison between the two methods is unfair due to the use of different dataset splits, as demonstrated by the drastically different baseline accuracies. Although it’s true that the proposed method achieves a proportionally larger improvement over the baseline compared with [Antoniou et al, 2017], the differences in experimental setup may be too large to draw a conclusion one way or the other about which method is better.

OVERALL
Although the paper is well-written and presents several intuitive methods for content/style decomposition with generative models, it’s hard to tell whether the results are significant due to incomplete comparison with prior work. On the generation side I would like to see a comparison especially with [Siddharth et al, 2017]. For low shot learning I think that the proposed method shows some promise, but it is difficult to draw hard conclusions from the experiments. For these reasons I lean slightly toward rejection.

MISSING REFERENCES
Siddharth et al, “Learning Disentangled Representations with Semi-Supervised Deep Generative Models”, NIPS 2017

Sohn, Lee, and Yan, “Learning structured output representation using deep conditional generative models”, NIPS 2015

Tzeng, Hoffman, Darrell, and Saenko, “Adversarial Discriminative Domain Adaptation”, CVPR 2017

---

> ### Author Response · Authors · 2018-11-26
> **Response**
>
> We thank the reviewer for their thorough evaluation and comments. In response to suggested comparisons with prior work, we have updated our paper with additional citations. However, the work cited by the reviewer does not address the problem we tackle. Our goal is to transfer style with open-ended content; Siddharth et al. [2017] and Tzeng et al. [2017] are concerned with a fixed set of content classes. Our goal is to perform content-style recombination, whereas Sohn et al. [2017] is primarily concerned with image completion and segmentation. While Tzeng et al. [2017] uses adversarial training via the GAN objective, it is otherwise not related to predictability minimization [Schmidhuber, 1992], which is a distinct adversarial method.
>
> Regarding the trade off of different losses: we explore varying the loss coefficients for CE, PM, and LF. These results are shown in Figure 8 in the Appendix. All components of STOC are needed to attain maximal performance.
>
> Regarding low-shot learning on Omniglot: We ran new simulations, included in the updated paper, showing the boost due to traditional data augmentation approaches (rigid image transformations), and showing a significant additional benefit of augmentation by synthetic examples. This puts our work on the same footing as the (still unpublished) DAGAN model by Antoniou et al. [2017], which includes some (undescribed) form of traditional data augmentation. With matched baselines, we perform comparably to DAGAN. DAGAN is designed specifically to perform data augmentation, whereas we are using data augmentation as a quantitative evaluation metric. DAGAN is unable to obtain the other results we report, such as recombination of content and style (Figures 1, 4, and 5), and does not explicitly perform content-style decomposition. Although DAGAN is also unpublished at present, we believe it’s valuable to have these two quite distinct methods in the literature as evidence suggesting intrinsic limitations to the benefit that can be obtained from synthetically generated examples.

---

### Official Review · AnonReviewer3 · 2018-11-03
**A novel method for content and style recombination in open domains**

**Rating:** 7
**Confidence:** 4

**Review:**

In this paper, the authors study an interesting problem called open-ended content style recombination, i.e., recombining the style of one image with the content of another image. In particular, the authors propose a VAE (variational autoencoder) based method (i.e., Style Transfer onto Open-Ended Content, STOC), which is optimized over a VAE reconstruction loss and/or a leakage filtering (LF) loss. More specifically, there are four variants of STOC, including CC (content classifier), CE (content encoding), PM (predictability minimization, Section 2.1) and LF (leakage filtering, Section 2.2). The main advantage of STOC is its ability to handle novel content from open domains. Experimental results on image synthesis and data set augmentation show the effectiveness of the proposed method in comparison with the state-of-the-art methods. The authors also study the comparative performance of four variants, i.e., CCF, CE, PM and LF.

Overall, the paper is well presented.

Some comments/suggestions:

(i) The authors are suggested to include an analysis of the time complexity of the proposed method (including the four variants).

(ii) The authors are suggested to include more results with different configurations such as that in Table 1 in order to make the results more convincing.

---

> ### Author Response · Authors · 2018-11-26
> **Response**
>
> We thank the reviewer for the thoughtful evaluation and feedback. Regarding the investigation of different model configurations, in the Appendix (Figure 8) we vary the coefficients of the various costs in the training objective function on the naturally-evaluated, synthetically-trained (NEST) task. We show that each component cost contributes to the model’s overall performance. Regarding time complexity, predictability minimization incorporates a GAN-like adversarial objective, which makes it strictly inferior to STOC in time complexity and--as we show in the paper--in the quality of synthesized images. The penalty in the STOC leakage filtering loss is proportional to the number of within- and between-class pairs that are drawn from P^+ and P^- in a minibatch.

---

### Meta-Review · Area_Chair1 · 2018-12-11
**decision**

**Confidence:** 4
**Recommendation:** Reject

**Metareview:**

The paper is on the borderline. From my reading, the paper presents a reasonable idea with quite good results on novel image generation and one-shot learning. On the other hand, the comparison against the prior work (both generation task and one-shot classification task) is not convincing. I also feel that there are many work with similar ideas (I listed some below, but these are not exhaustive/comprehensive list), but they are not cited or compared, I am not sure about if the proposed concept is novel in high-level. Although some implementation details of this method may provide advantages over other related work, such comparison is not clear to me.

Disentangling factors of variation in deep representations using adversarial training
https://arxiv.org/abs/1611.03383
NIPS 2016

Rethinking Style and Content Disentanglement in Variational Autoencoders
https://openreview.net/forum?id=B1rQtwJDG
ICLR 2018 workshop

Disentangling Factors of Variation by Mixing Them
http://openaccess.thecvf.com/content_cvpr_2018/papers/Hu_Disentangling_Factors_of_CVPR_2018_paper.pdf
CVPR 2018

Separating Style and Content for Generalized Style Transfer
https://arxiv.org/pdf/1711.06454.pdf

Finally, I feel that the writing needs improvement. Although the method is intuitive and has simple idea, the paper seems to lack full details (e.g., principled derivation of the model as a variant of the VAE formulation) and precise definitions of terms (e.g., second term of LF loss).